# Long-Term Multilocal Monitoring of Leaf Rust Resistance in the Spring Bread Wheat Genetic Resources from Institute of Plant Genetic Resources (VIR)

Lev Gennadievich Tyryshkin [1,*][ID], Yuliya Vital'evna Zeleneva [2][ID], Alla Nikolaevna Brykova [1], Evgeniya Yurievna Kudryavtseva [1], Valentina Alekseevna Loseva [3], Magomed Alievich Akhmedov [4], Asef Zilfikarovich Shikhmuradov [4] and Evgeny Valerievich Zuev [1]

1   N.I. Vavilov All-Russian Institute of Plant Genetic Resources, 42, 44 Bolshaya Morskaya Street, 190000 St. Petersburg, Russia; a.brykova@vir.nw.ru (A.N.B.); f-evgenya@rambler.ru (E.Y.K.); e.zuev@vir.nw.ru (E.V.Z.)
2   All-Russian Institute of Plant Protection, Podbel'skogo Street, 3, 196608 St. Petersburg, Russia; zelenewa@mail.ru
3   Yekaterinino Experimental Station, Branch of the N.I. Vavilov Institute of Plant Genetic Resources (VIR), s. Yekaterinino, Nikiforovskii District, 393023 Tambov, Russia; a.mordasowa2014@yandex.ru
4   Dagestan Experimental Station, Branch of the N.I. Vavilov Institute of Plant Genetic Resources (VIR), s. Vavilovo, 368612 Derbent, Russia; makhmed.dos@mail.ru (M.A.A.); asef121263@mail.ru (A.Z.S.)
*   Correspondence: tyryshkinlev@rambler.ru

**Abstract:** *Triticum aestivum* L. (bread wheat) is the most important cereal crop in world grain production, including in the territory of the Russian Federation. One of the most important factors influencing the yield and quality of wheat grain is the affection of plants with leaf rust (*Puccinia triticina* Erikss.). To broaden the set of sources for effective rust resistance, spring bread wheat samples from N.I. Vavilov All-Russian Institute of Plant Genetic Resources (VIR) were monitored for adult resistance to the disease under natural infections for many decades at three distant locations of the Russian Federation: the Dagestan Experimental Station (DES) of VIR (10,549 accessions), Yekaterinino Experimental Station (YES) (4384 accessions), and Pushkin Experimental Field (PEF) (7704 accessions). Information on the disease development at these three stations is presented at least for 51 last years. As a result of disease development evaluation under natural epiphytotic conditions for not less than 3 years, 293 (15 landraces, 127 breeding lines, and 151 commercial varieties), 118 (1 landrace, 38 breeding lines, and 79 commercial varieties), and 127 (10 landraces, 48 breeding lines, and 69 commercial varieties) samples were classified as resistant to leaf rust at DES, YES, and PEF, respectively. Among samples from the State Register of Breeding Achievements in Russia, 15, 13, and 8 spring wheat varieties were resistant to leaf rust at DES, YES, and PEF, respectively. Juvenile resistance was estimated under laboratory conditions after seedling inoculation with a complex population of *P. triticina*: 73 highly resistant varieties and breeding lines were identified; all landraces, including those classified as resistant in the fields, were susceptible to disease at the seedling stage. A total of 26 wheat accessions were identified to be resistant to leaf rust at two to three locations; 14 of them possess adult resistance, and 12 samples have seedling resistance. According to results of PCR amplification with primers specific to markers of effective genes for leaf rust resistance, 6 accessions have gene *Lr9*, 1 sample is protected by *Lr19*, and 1 sample possesses gene *Lr24*. Wheat samples identified as possessing effective seedling or adult resistance could be of interest for breeding in some regions of the Russian Federation and other countries.

**Keywords:** bread wheat; leaf rust; adult resistance; seedling resistance; genes for resistance; long-term multilocal monitoring

## 1. Introduction

*Triticum aestivum* L. (bread wheat) is the most important cereal crop in world grain production, including in the territory of the Russian Federation. The area for wheat growing in Russia in 2020 was 28.7 million hectares [1]; the total grain yield of crop this year amounted to 85,896 million tonnes [2].

One of the most important factors influencing the yield and quality of wheat grain is the affection of plants with fungal leaf diseases. One of them is leaf rust (*Puccinia triticina* Erikss.); it is a harmful disease in all wheat-growing regions. Epiphytoties of the disease are observed every 2–3 years with crop losses in commercial varieties estimated to be at 20–40% [3]. Despite the development of many methods of controlling rust, it is generally accepted that the best of them, in terms of both economic efficiency and environmental impact, is the cultivation of resistant varieties. The most important step in the creation of such varieties is the search for new sources for effective resistance. Currently, among the varieties of spring wheat allowed for growing in the Russian Federation, there is quite a high frequency of genotypes with juvenile (seedling) and adult resistance to leaf rust, but all of them are protected by an extremely limited number of effective genes [4]. Their extensive use in wheat breeding will lead to their effectiveness loss due to microevolution-ary processes in the rust pathogen. As a result, the search for new resistant material is of high interest especially with new effective genes for resistance that has never been used in wheat breeding. The collection of spring bread wheat preserved in N.I. Vavilov All-Russian Institute of Plant Genetic Resources (VIR) (Department of Wheat Genetic Resources) now consists of 15,048 samples from 97 countries. The most represented are accessions from Russia, Mexico, India, Kazakhstan, China, Australia, the USA, Tajikistan, Pakistan, and Turkey. The first specimens were included in the collection as early as in 1907. Most of the samples were introduced in the 1970s. Since 1992, the Russian stage of the collection formation has begun. The up-to-date VIR collection includes landraces, breeding varieties, breeding lines, and genetic lines. Local varieties compose 40% of the collection, and 60% of the samples are breeding material. A study on effective resistance to leaf rust in a vast plant gene pool under field and laboratory conditions can lead to the broadening of genetic diversity for a trait in newly developed wheat varieties and hence to the decreasing of yield losses from disease.

The Department of Wheat Genetic Resources has been constantly creating databases for results of field wheat sample evaluations at experimental stations of the institute. Here, we present the results of a long-term leaf rust resistance study on spring wheat samples from the VIR collection at three experimental stations and the results of seedling resistance evaluation in a broad set of crop samples.

## 2. Environments, Materials, and Methods

*2.1. Regions of Wheat Assessment for Leaf Rust Resistance*

Leaf rust development on wheat was assessed at three VIR experimental stations (Figure 1).

The Dagestan Experimental Station (DES) of a VIR branch is located in the Vavilovo village near the town of Derbent (Republic of Dagestan, North Caucasus Region of Russia). The type of soil at the station is light chestnut soil. Spring wheat is grown under irrigation. The climate is transitional from moderate to subtropical semidry. Weather conditions are affected by the Caspian Sea, so autumn is long and warm, and spring comes with a delay. Winter is mild, snow lays only 2 weeks a year. Summer is long and hot. The average annual temperature in Derbent is positive: +12.5 °C, the average monthly temperature in January is +1.4 °C, and the average monthly temperature in July is +24.6 °C. The duration of the warm period is 270 days in a year. The average rainfall quantity is 363 mm per year; the most rains are in October. The average annual relative humidity is 78.8%. High humidity and temperatures in the range of +16–22 °C during May–June (heading–wheat maturation) result in severe leaf rust development in many years [5].

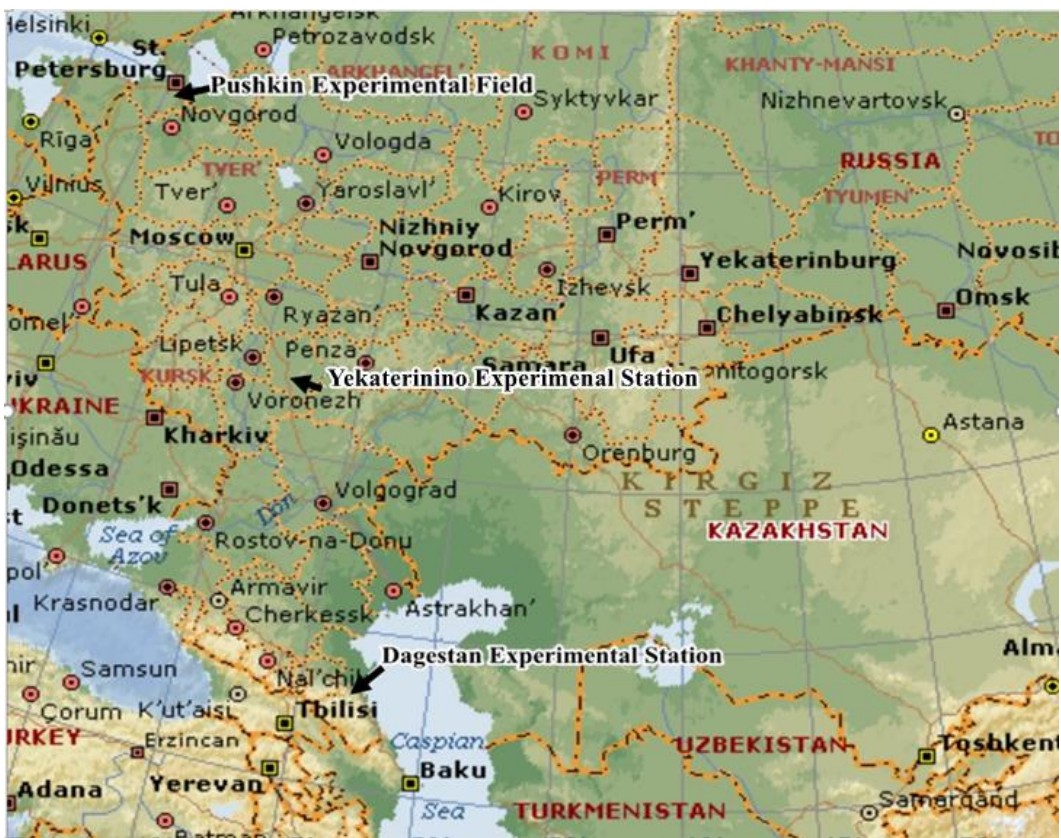

**Figure 1.** Locations of evaluation experiments of spring wheat germplasm resistance to leaf rust.

The Yekaterinino Experimental Station branch of VIR (YES) is located in the Yekaterinino village (Nikiforovo District, Tambov Region, Central Black Earth Region of Russia). The soils at the station are chernozem. Spring wheat is grown under the absence of artificial irrigation. The climate is moderately continental, with cold and long winters and warm, moderately dry summers. The duration of the period with a temperature above 10 °C is 145 days, and the temperature sum at this period reaches 2500 °C. The average air temperature is positive—+4.3 °C. The lowest average monthly temperature in January is −10.8 °C, and the highest average monthly temperature in July is +20.0 °C. The average annual precipitation amount is 480–500 mm. The amount of precipitation for the period with an average air temperature above 10 °C is 230–265 mm. Temperatures in June–July, +18–20 °C, are optimal for the development of leaf rust, but the moisture deficiency in some years leads to the suppression of pathogen development [6].

The Pushkin Experimental Field (PEF) is a part of the Pushkin and Pavlovsk Laboratories of VIR (Russia, Northwest Region). The soil at the field is podzolic. Wheat is grown under the absence of artificial irrigation. The climate of the Pushkin District of Saint Petersburg is moderate, transitioning from sea to continental. The area is characterized by high humidity and changeable and cloudy weather with moderately cold winters and mild summers. The average air temperature is positive—+5.2 °C. The lowest average monthly temperature in January is −5.6 °C, and the highest average monthly temperature in July is +19.1 °C. During spring wheat heading and maturation (July–August), the average air temperature varies from +17.8 to 16 °C. Precipitation during the growing season is 251 mm. Weather conditions favor the development of leaf rust on wheat crops [7].

*2.2. Plant Material*

The materials of the study were samples of spring bread wheat from the VIR collection; the origin of the samples studied for resistance to leaf rust at three experimental stations of

the institute is presented in Table 1. The countries of origin of the samples are united in the geographical regions of the world according to methods accepted in VIR [8].

**Table 1.** Origin of spring wheat samples studied for leaf rust resistance in the VIR system.

| Region | Number of Samples Evaluated at | | |
|---|---|---|---|
| | **DES** | **YES** | **PEF** |
| Russia | 1376 | 1217 | 2100 |
| Europe | 1226 | 802 | 1465 |
| Transcaucasia | 326 | 9 | 33 |
| Small Asia and the Middle East | 522 | 43 | 98 |
| Central Asia | 2811 | 683 | 884 |
| Southwest and East Asia | 571 | 112 | 413 |
| Africa | 635 | 129 | 227 |
| North and Central America | 1629 | 636 | 1604 |
| South America | 883 | 524 | 512 |
| Australia and New Zealand | 557 | 219 | 356 |
| Unknown | 13 | - | 12 |
| **Total** | **10,549** | **4384** | **7704** |
| % Whole collection | 70.1 | 29.1 | 51.2 |

At the DES, resistance to leaf rust was studied under field conditions in 10,549 samples (70.1% of the entire collection of spring wheat) from 1962 up to 2020. The largest numbers of entries were from Russia (1376), Mexico (1113), India (692), Australia (545), and Tajikistan (488). Countries such as Mali and Chad were represented with single samples.

At the YES, field evaluation of resistance to rust was conducted in 1970–2020; a total of 4384 varieties and lines (29.1% of the entire collection) (Table 1) were studied. The largest numbers of samples were from the same countries as in the DES study. Mali and Chad were also presented as single samples.

Field assessment of leaf rust resistance at the PEF was conducted from 1945 till 2020 with 7704 samples of spring wheat (51.2% of the entire collection) (Table 1). The largest numbers of samples originated from Russia (2100), Mexico (966), the USA (412), Australia (347), Kazakhstan (346), China (325), Germany (273), and Canada (226). To a lesser extent, the countries of Guatemala, Jordan, Libya, Mali, Mauritania, Paraguay, Syria, and Zimbabwe were represented.

### 2.3. Methods

Field evaluations of resistance were conducted at the natural development of leaf rust using the conventional agricultural technique for spring bread wheat cultivation. The seeding of samples was carried out at the optimal time for a certain region in plots of 1 m$^2$. The standard susceptible varieties were Diamant, Leningradka, Leningradskaya 97 and Leningradskaya 6 (PEF), Siete Cerros (DES), and Kutulukskaya (EOS). Standard varieties were sown after every 20 experimental samples.

The disease ratings were scored twice during the growing season at the heading and milky stages. Classification of a sample for leaf rust resistance was performed according to a 1–9 scale [9–11] (Table 2).

The degree of disease development in a particular year of the study was determined by the average development of leaf rust on standards throughout the field. Disease ratings of 1–3 on standard varieties corresponded to high, and 3–5 to moderate leaf rust development. To identify reliably resistant samples, only data for 3 or more years of high and moderate disease development were taken into analysis.

**Table 2.** Scale for field assessment of wheat for leaf rust resistance.

| Disease Rating | Level of Resistance | Pustules | Leaf Surface Affected, % |
|---|---|---|---|
| 9 | Very high | Single, very small, surrounded with necrosis | 0–5 |
| 7 | High | Small, sometimes surrounded with chlorosis | <10 |
| 5 | Moderate | Small, without or with chlorosis | ~20 |
| 3 | Low | Large, sometimes merging, especially abundant at middle leaves | ~50 |
| 1 | Very low | Large, dense, merging at middle and sometimes upper leaves | >50 |

A whole set of spring wheat entries to the VIR collection since 2000 was evaluated for seedling (juvenile) resistance to leaf rust. A complex population of *P. triticina* was created by combining samplings from leaves of several susceptible wheat varieties in the Northwest Region of Russia and North Caucasus and was maintained in the laboratory for seedlings of cv. Leningradka in a light chamber (light 2500 lux, temperature—20–22 °C). Under these conditions, the population was virulent/avirulent to lines and samples with resistance genes *Lr1, 2a, 2c, 3bg, 10, 11, 12, 13, 14a, 14b, 15, 16, 17, 18, 20, 21, 22a, 22b, 23, 25, 26, 28, 29, 27 + 31, 32, 33, 34, 35, 36, 37, 38, 43, 44, 45, 46, 48, 49, 52, 57, 60, 64/Lr9, 19, 24, 39 (=41)*, and *47*. Water suspension of the fungus uredospores ($30 \times 10^3$ spores/mL) in water was used to infect the experimental material. To study juvenile resistance to the disease, 10–15 seeds of each sample were placed on cotton wool rolls wetted with water. After 10–15 days, seedlings at the stage of 1–2 leaves were placed in cuvettes horizontally and sprayed with water suspensions of uredospores with the use of a hand atomizer. After inoculation, the cuvettes were wrapped with polyethylene, covered with glass, and placed in the darkness. A day later, the polyethylene film and glass were removed, and seedlings were returned to a vertical position [4].

The types of reaction to *P. triticina* infection were scored on the 12th day after inoculation according to a generally accepted scale [12] with some modifications, where: 0—no disease symptoms; 0—necrotic spots without pustules; 1—very small pustules surrounded by necrosis; 2—medium-sized pustules surrounded by necrosis or chlorosis; 3—large pustules without necrosis; s.p.—single pustules of a susceptible type without necrosis; X—pustules of different types on one leaf. Types 0, 0, and 1 correspond to a high level of resistance; 2, s.p., and X to a moderate level of resistance; and 3 to susceptibility.

Samples resistant to leaf rust in the fields at two to three experimental stations and possessing resistance at seedling stages of ontogenesis were used to identify effective genes for the trait with the use of DNA markers tightly linked to genes *Lr9*, *Lr19*, and *Lr24*. DNA was extracted from three seedlings with a micromethod proposed by Edwards et al. [13] with modifications by Dorokhov and Klocke [14]. The DNA concentration was 50 ng/μL. The markers of identified genes are presented in Table 3. Polymerase chain reaction (PCR) was performed in a thermocycler, C-1000 (Bio-Rad, Hercules, CA, USA), according to original protocols [15–17]. Amplification products were electrophoresed on 1.5% agarose gel in 1 × Tris–boric acid–EDTA (TBE) buffer.

**Table 3.** Allele-specific primers of wheat genes for effective leaf rust resistance.

| Gene | Marker | Nucleotide Sequences of Primers | Size of DNA Amplification Product, bp | Reference |
|---|---|---|---|---|
| *Lr9* | SCS5 | F: 5′-TGC GCC CTT CAA AGG AAG-3′<br>R: 5′-TGC GCC CTT CTG AAC TGT AT-3′ | 550 | [15] |
| *Lr19* | SCS265 | F: 5′-GGC GGA TAA GCA GAG CAG AG-3′<br>R: 5′-GGC GGA TAA GTG GGT TAT GG-3′ | 512 | [16] |
| *Lr24* | Sr24≠12 | F: 5′-CAC CCG TGA CAT GCT CGT A-3′<br>R: 5′-AAC AGG AAA TGA GCA ACG ATG T-3′ | 500 | [17] |

## 3. Results

### 3.1. Resistance of Wheat Samples under the Conditions of the Dagestan Experimental Station of VIR

High and moderate levels of leaf rust development in the field of this station were observed in 41 seasons of spring wheat vegetation out of 59 years of the study (Table 4).

**Table 4.** Leaf rust development at the Dagestan Experimental Station of VIR.

| Year | Disease Development | Year | Disease Development | Year | Disease Development | Year | Disease Development |
|---|---|---|---|---|---|---|---|
| 1962 | high | 1977 | high | 1992 | moderate | 2007 | high |
| 1963 | high | 1978 | high | 1993 | moderate | 2008 | high |
| 1964 | high | 1979 | moderate | 1994 | very low | 2009 | very low |
| 1965 | high | 1980 | high | 1995 | moderate | 2010 | very low |
| 1966 | moderate | 1981 | moderate | 1996 | low | 2011 | very low |
| 1967 | moderate | 1982 | moderate | 1997 | low | 2012 | moderate |
| 1968 | high | 1983 | moderate | 1998 | moderate | 2013 | very low |
| 1969 | high | 1984 | high | 1999 | low | 2014 | moderate |
| 1970 | moderate | 1985 | moderate | 2000 | very low | 2015 | very low |
| 1971 | very low | 1986 | moderate | 2001 | high | 2016 | moderate |
| 1972 | very low | 1987 | low | 2002 | very low | 2017 | moderate |
| 1973 | moderate | 1988 | moderate | 2003 | low | 2018 | very low |
| 1974 | high | 1989 | moderate | 2004 | moderate | 2019 | low |
| 1975 | high | 1990 | moderate | 2005 | moderate | 2020 | low |
| 1976 | moderate | 1991 | high | 2006 | moderate | | |

According to multiannual results, screenings of 34 and 259 samples were highly resistant and resistant to leaf rust, respectively. The greatest number of resistant genotypes originated from the USA (35), Russia (34), Mexico (29), and Australia (25); their distribution among world regions is shown in Figure 2. Highly resistant and resistant samples are represented as landraces and breeding material (Figure 3).

Among landraces, the next samples were classified as resistant to the disease: Stromberg Rooi (k-5717, South Africa), k-11900 (Georgia), A1 (k-28752, China), k-29867 (Argentina), k-34482 (Kazakhstan), k-34679 (Kazakhstan), k-36730 (Kazakhstan), Palvan-Bugdai (k-51056, Uzbekistan), k-55839 (Tunisia), k-55844 (Tunisia), k-60211 (Yemen), Local 2427 (k-63081, Egypt), Local 783 (13) (k-63082, Egypt), Local 799 (k-63083, Egypt), k-63085 (Egypt).

The resistant varieties from the State Register of Russian Federation (RF) were Kinel'skaya 60 (k-62643, Samara Region), Altaiskaya 110 (k-65128, Altai Region), Gerkules (k-65129, Omsk Region), Maria 1 (k-65130, Kemerovo Region), Pamyati Vavenkova (k-65132, Novosibirsk Region), Pamyati Aphrodity (to-65135, Kemerovo Region), Svirel (k-65136, Krasnoyarsk Region), Saratovskaya 74 (k-65139, Saratov Region), Raduga (k-65240, Kurgan Region), Tulaikovskaya 100 (k-64643, Samara Region), Kinel'skaya Niva (k-64666, Samara Region), Novosibirskaya 44 (k-64867, Novosibirsk Region), Chelyaba Stepnaya (k-64872, Chelyabinsk Region), Voevoda (k-64997, Saratov Region), and Favorit (k-64998, Saratov region).

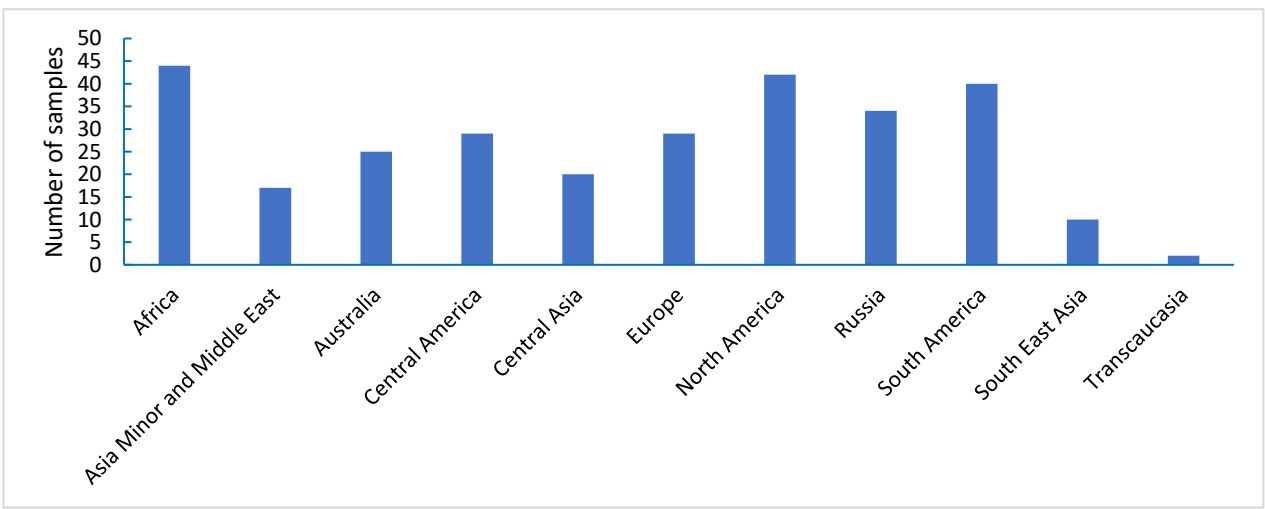

**Figure 2.** Number of spring bread wheat samples resistant and highly resistant to leaf rust under the conditions of the Dagestan Experimental Station of VIR.

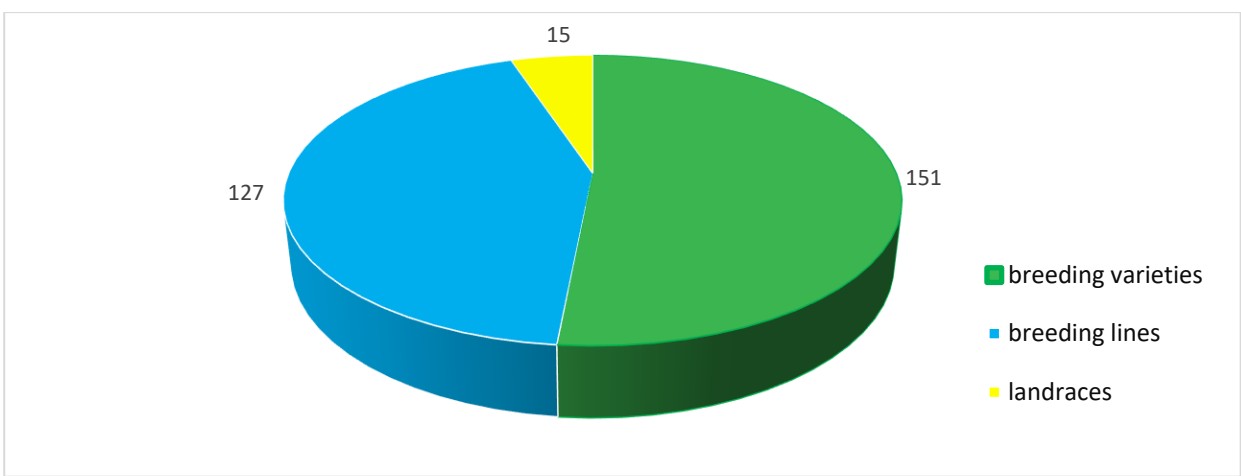

**Figure 3.** Distribution of spring wheat samples resistant and highly resistant to leaf rust at the Dagestan Experimental Station of VIR for their status.

*3.2. Resistance of Wheat Samples under the Conditions of the Yekaterinino Experimental Station of VIR*

At the YES, wheat leaf rust was observed in high and moderate degrees in 25 and 12 years, respectively, out of 51 when accessions under study were grown. In the last years at this station, disease development was significant annually except in 2015, 2018, and 2019, when rust on bread spring wheat was not recorded (Table 5).

According to the results of the multiyear evaluations, 79 samples were resistant to the disease and 39 samples were highly resistant to rust. The largest numbers of resistant samples originated from Russia (25), Brazil (23), and Mexico (12). The distribution of the identified resistant samples in world regions is presented in Figure 4. Almost all the accessions were represented by breeding material (79 varieties and 38 lines), with the exception of one local variety (landrace) from Yemen—k-58202.

Out of the Russian commercial varieties, L 503 (k-60620, Saratov Region), Duet (k-63500, Chelyabinsk Region), Tulaikovskaya Zolotistaya (k-63715, Samara Region), Aria (k-64545, Kurgan Region), Chelaba Yubileinaya (k-64694, Chelyabinsk Region), and Sibirskii Al'yans (k-65242, Altai Region) were classified as resistant, and Tulaikovskaya 5 (k-62927, Samara Region), Tulaikovskaya 100 (k-64643, Samara Region), Kinel'skaya Niva (k-64666, Samara Region), Novosibirskaya 44 (k-64867, Novosibirsk Region), Stepnaya Chelyaba

(k-64872, Chelyabinsk Region), Voevoda (k-64997, Saratov Region), and Favorit (k-64998, Saratov Region) were highly resistant to leaf rust.

**Table 5.** Leaf rust development at the Yekaterinino Experimental Station of VIR.

| Year | Disease Development | Year | Disease Development | Year | Disease Development | Year | Disease Development |
|---|---|---|---|---|---|---|---|
| 1970 | moderate | 1983 | high | 1996 | low | 2009 | high |
| 1971 | low | 1984 | high | 1997 | moderate | 2010 | low |
| 1972 | low | 1985 | high | 1998 | high | 2011 | high |
| 1973 | low | 1986 | low | 1999 | moderate | 2012 | moderate |
| 1974 | low | 1987 | high | 2000 | moderate | 2013 | high |
| 1975 | high | 1988 | low | 2001 | moderate | 2014 | moderate |
| 1976 | low | 1989 | high | 2002 | moderate | 2015 | low |
| 1977 | moderate | 1990 | high | 2003 | high | 2016 | moderate |
| 1978 | moderate | 1991 | high | 2004 | high | 2017 | high |
| 1979 | low | 1992 | high | 2005 | high | 2018 | low |
| 1980 | high | 1993 | high | 2006 | high | 2019 | low |
| 1981 | low | 1994 | high | 2007 | high | 2020 | moderate |
| 1982 | high | 1995 | high | 2008 | high | | |

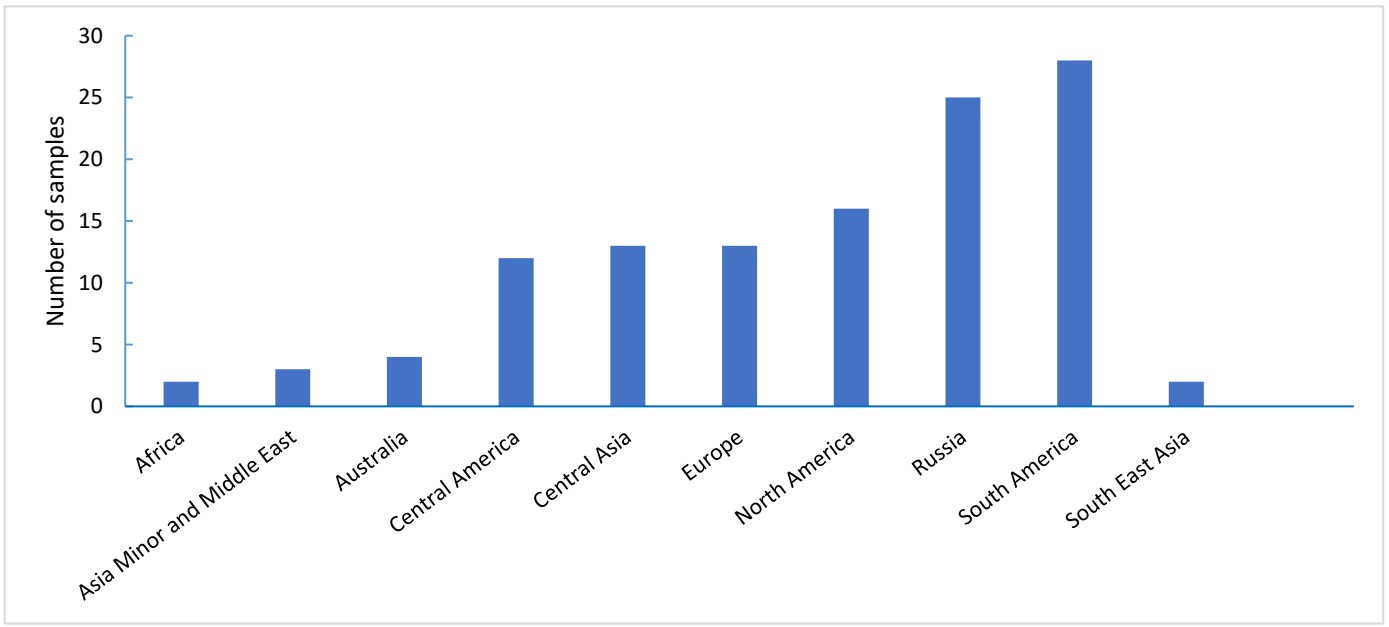

**Figure 4.** Number of spring bread wheat samples resistant and highly resistant to leaf rust identified at the Yekaterinino Experimental Station of VIR.

*3.3. Resistance of Wheat Samples under the Conditions of the VIR Pushkin Experimental Field*

Under Pushkin town conditions, the development of wheat leaf rust was high and moderate in 12 and 29 years, respectively, out of 51 years under study (Table 6).

According to the results of multiyear evaluations, 90 samples were resistant to the disease and 37 samples were highly resistant. The largest numbers of resistant forms are from the United States (21), Canada (13), Argentina (13), Mexico (10), and Russia (9); the distribution of identified samples in world regions is presented in Figure 5. Resistant and highly resistant accessions were presented by both local varieties and breeding material (Figure 6).

**Table 6.** Leaf rust development on wheat at the Pushkin Experimental Field.

| Year | Development of the Disease | Year | Development of the Disease | Year | Development of the Disease | Year | Development of the Disease |
|---|---|---|---|---|---|---|---|
| 1945 | moderate | 1968 | moderate | 1985 | moderate | 2001 | moderate |
| 1947 | moderate | 1969 | moderate | 1986 | high | 2002 | moderate |
| 1948 | moderate | 1970 | low | 1987 | low | 2004 | low |
| 1951 | moderate | 1971 | moderate | 1988 | moderate | 2006 | low |
| 1957 | high | 1972 | moderate | 1989 | high | 2007 | low |
| 1959 | moderate | 1973 | moderate | 1991 | moderate | 2009 | moderate |
| 1961 | high | 1974 | moderate | 1992 | low | 2010 | high |
| 1962 | moderate | 1975 | low | 1993 | low | 2013 | moderate |
| 1963 | high | 1980 | high | 1994 | moderate | 2014 | high |
| 1964 | high | 1981 | moderate | 1995 | moderate | 2017 | moderate |
| 1965 | moderate | 1982 | moderate | 1996 | low | 2019 | high |
| 1966 | high | 1983 | moderate | 1999 | moderate | 2020 | low |
| 1967 | moderate | 1984 | moderate | 2000 | high | | |

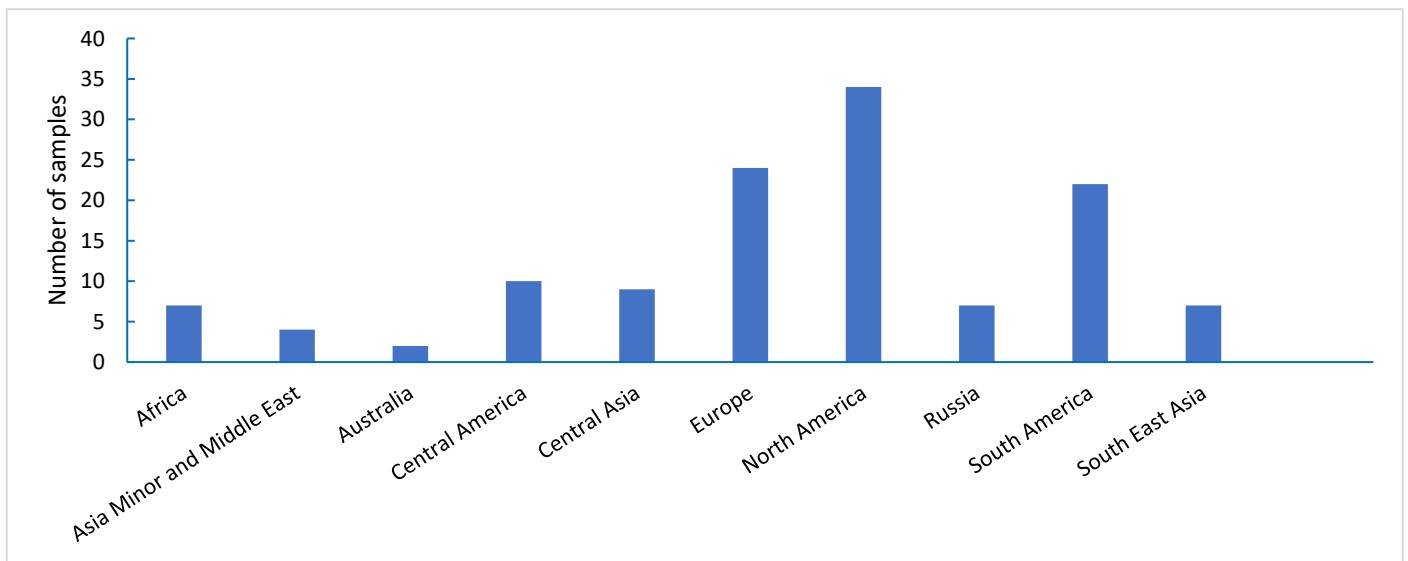

**Figure 5.** Distribution of wheat samples resistant and highly resistant to leaf rust at the Pushkin Experimental Field.

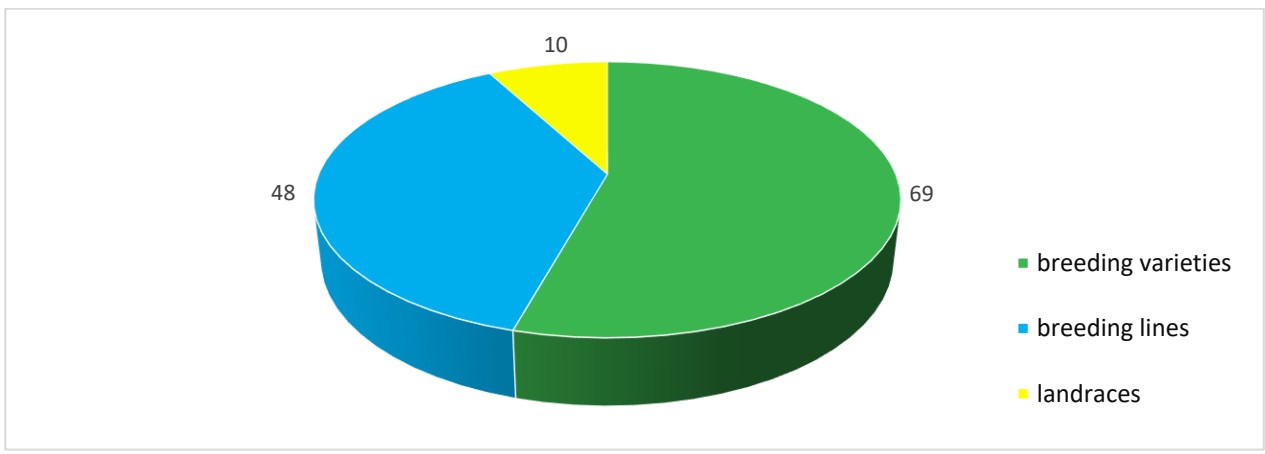

**Figure 6.** Distribution of spring wheat samples resistant and highly resistant to leaf rust at the Pushkin Experimental Field for their status.

Among the local samples (landraces) resistant to leaf rust were k-7895 (Russia), 137 (k-28352, China), O1(a) (k-28667, China), Ob(c) (k-28681, China), A1 (k-28752, China), k-29867 (Argentina), k-39538 (Pakistan), Iskamish-K-2-Dark (k-55127, Afghanistan), k-55861 (Portugal), and k-55863 (Portugal).

Varieties from the Russian registry—Kinel'skaya 60 (k-62643, Samara Region), Prokhorovka (k-62644, Saratov Oblast), Tulaikovskaya 10 (k-63714, Samara Region), Tulaikovskaya Zolotistaya (k-63715, Samara Region), Boyevchanka (k-64983, Omsk Region), Kinel'skaya Yubileinaya (k-66270, Samara Region), KWS Torridon (k-66273, United Kingdom), and Arhat (k-66406, Penza Region)—were resistant to leaf rust at this point of the evaluation.

### 3.4. Juvenile Resistance of Wheat Samples to Leaf Rust

In 1999–2020, 3809 samples of spring bread wheat were evaluated for seedling resistance to the disease. A total of 73 highly resistant varieties and breeding lines were identified. Most of these samples were from Russia (Table 7). All landraces, including those classified as resistant in the fields at three experimental stations, were highly susceptible to the disease at the seedling stage.

Due to the fact that juvenile-resistant samples are of greatest interest for breeding purposes, we present a full list of the identified accessions (Table 7).

According to results of molecular marking of 12 samples resistant to leaf rust in the fields at two to three experimental stations and possessing resistance at seedling stages of ontogenesis, gene *Lr19* was identified in cultivar Kinel'skaya Niva, *Lr24* in cv. Tasman, and gene *Lr9*—in samples of Novosibirskaya 44, Chelyaba Stepnaya, Lutescens 30, ANK-4, Olga, and Lubninka (Figures 7–9).

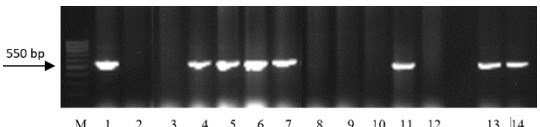

**Figure 7.** Amplification products after PCR with primers to SCS5, linked to gene *Lr9*: M—100-bp DNA ladder, 1—ANK-4, 2—LT-1, 3—Favorit, 4—Lubninka, 5—Novosibirskaya 44, 6—Cheliaba Stepnaya, 7—Olga, 8—Tasman, 9—Kinelskaya Niva, 10—Voevoda, 11—Lutescens 30, 12—Tulaikovskaya 100, 13, 14—Thatcher *Lr9*.

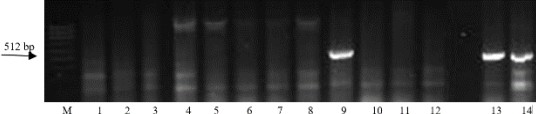

**Figure 8.** Amplification products after PCR with primers to SCS265, linked to gene *Lr19*: M—100-bp DNA ladder, 1—ANK-4, 2—LT-1, 3—Favorit, 4—Lubninka, 5—Novosibirskaya 44, 6—Cheliaba Stepnaya, 7—Olga, 8—Tasman, 9—Kinelskaya Niva, 10—Voevoda, 11—Lutescens 30, 12—Tulaikovskaya 100, 13, 14—Thatcher *Lr19*.

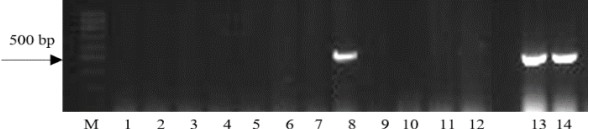

**Figure 9.** Amplification products after PCR with primers to Sr24≠12, linked to gene *Lr24*: M—100-bp DNA ladder, 1—ANK-4, 2—LT-1, 3—Favorit, 4—Lubninka, 5—Novosibirskaya 44, 6—Cheliaba Stepnaya, 7—Olga, 8—Tasman, 9—Kinelskaya Niva, 10—Voevoda, 11—Lutescens 30, 12—Tulaikovskaya 100, 13, 14—Thatcher *Lr24*.

**Table 7.** Origin of spring wheat samples with effective juvenile resistance to leaf rust.

| Country/Region | Accessions |
|---|---|
| Australia | Skua (k-60613), Tasman (k-63180), Cunningham (k-64209), Sunstate (k-64218) |
| Brazil | OCEPAR 11—Juriti (k-62612), OCEPAR 8—Macuco (k-62614), BR 31—Miriti (k-62619), CEP 14—Tapes (k-60944), BR 34 (k-62185) |
| Canada | AC Barrie (k-64596) |
| Mexico | k-65603 |
| Netherlands | Tybalt (k-64897) |
| Poland | Nawra (k-64708) |
| South Africa | SST-23 (k-64138), SST-25 (k-64140) |
| Sweden | WW 17,283 (k-60997) |
| Switzerland | Toronit (k-66032) |
| United Kingdom | Sparrow (k-66090) |
| USA | Stoa (k-59033), Wampum (k-60588), MN 81,330 (k-60785), Russ (k-64595) |
| Unknown | PS 131 (k-64597), PS 133 (k-64598) |
| Russia, Altai | Altaiskaya 65 (k-64455), Altaiskaya 110 (k-65128) |
| Russia, Chelyabinsk | Quinta (k-63467), Duet (k-63500), Pamiaty Ruba (k-64378), Chelyaba 2 (to-64379), Chelyaba 75 (k-64871), Chelyaba Stepnaya (k-64872) |
| Russia, Kemerovo | Tuleevskaya k-63461) |
| Russia, Kurgan | Aria (k-64545) |
| Russia, Leningrad | LT 1 (k-65816) |
| Russia, Novosibirsk | ANK-4 (k-56395), Obskaya 14 (k-64363), Udacha (k-64372), Sibirskaya 24 (k-66442), Lubninka (k-64866), Novosibirskaya 44 (k-64867), Olga (k-65000), Novosibirskaya 18 (k-65820), Sibirskaya 17 (k-66017), Sibirskaya 21 (k-66269) |
| Russia, Omsk | Lavrusha (k-64984), Omskaya 41 (k-65253), OmGAU 100 (k-66387) |
| Russia, Penza | Yulia (k-63717); |
| Russia, Samara | Tulaikovskaya 10 (k-63714), Tulaikovskaya Zolotistaya (k-63715), Tulaikovskaya 100 (k-64643), Lutescens 30 (k-64647), Lutescens 101 (k-64648), Lutescens 13 (k-64649), Kinel'skaya Niva (k-64666), Tulaikovskaya 108 (k-65452), Ekada 113 (k-65453), Tulaikovskaya 110 (k-65454), Tulaikovskaya Nadezhda (k-65827), Kinel'skaya Yubileinaya (k-66270), Kinel'skaya Volna (k-66274), Tulaikovskaya 116 (k-66347); |
| Russia, Saratov | L 505 (k-62892), Dobrynya (k-64252), Voevoda (k-64997), Favorit (k-64998), Lebedushka (k-66410) |
| Russia, Tambov | Mertsana (k-65449) |
| Russia, Tyumen | Latona (k-64359), Tyumenka (k-66271); |
| Russia, Ulyanovsk | Ulyanovskaya 105 (k-66011) |
| Russia, Voronezh | Voronezhskaya 18 (k-65998), Voronezhskaya 20 (k-66257) |

## 4. Discussion

Wheat leaf rust is one of the widespread and harmful diseases of bread wheat *Triticum aestivum* L., decreasing the yield and quality of the crop. Despite the development of quite a wide range of methods to protect the crop from the disease, the creation of resistant varieties is considered to be the most profitable and environmentally friendly one. To develop these varieties, the search for sources of new effective genes for resistance remains a very important task. So far, 77 genes for wheat leaf resistance [18,19] have been found, but most of them are ineffective in populations of *P. triticina* from Russia [20,21]. At the juvenile stage, only genes *Lr9*, *19*, *24*, *41*, and *47* were highly effective against the pathogen population from the Northwest Region of the Russian Federation [22], but the first three

of them have already lost their effectiveness in some regions of the country [23–25]. In the past, a large quantity of wheat samples resistant to leaf rust at the adult stage were identified in the world [26–36], but evidently, they cannot be a priori regarded as valuable for breeding in the Russian Federation due to both resistance dependence on the pathogen population structure for virulence and specific for certain region environmental conditions.

The world collection of N.I. Vavilov All-Russian Institute of Plant Genetic Resources (VIR, Russia, Saint Petersburg) is of great interest for the search for valuable breeding material, including wheat for resistance to harmful foliar diseases. It consists of 15,048 accessions of bread spring samples from 97 countries. It presents landraces, breeding, and genetic lines, and varieties.

Systematic field evaluations of wheat VIR collection samples for leaf rust resistance have been initiated since as early as after World War II.

Every year, new entries of wheat accessions from the collection are sent to VIR's experimental stations for 3 years of field studies. The results of the studies are fragmentarily published in editions of the VIR World Collection Catalogues in Russian. The most informative data for leaf rust development evaluation at PEF, DES, and YES were published with 2750, 799, and 228 samples, respectively [37–42]. Data for seedling resistance were published with 1961 samples [43–47].

Due to the variability of phytopathogen populations for virulence and aggressiveness at the same area in different years and the dependence of phenotypical resistance expression on environmental factors, reliable data on resistant sample identification can only be obtained by studying the trait for 3 or more epiphytotic years [48] Therefore, in this paper we classified as resistant only accessions showing scores of 7 and 9 at each station if they were grown during not less than 3 years of high to moderate development of the rust disease.

As a result of long-term multilocal monitoring of leaf rust resistance in spring bread wheat accessions from the VIR collection, we identified 293, 118, and 127 highly resistant and resistant samples under the field conditions of DES, YES, and PEF, respectively. The largest number of resistant accessions was found at DES because the largest set of plant materials was sown there (10,549); it is because the Dagestan Experimental Station was founded by N.I. Vavilov especially for cereal immunological studies, and all entries into the collection were evaluated first of all at this station [49].

Different reasons could be responsible for the noncoincidence of identified resistant samples at three 3 VIR stations. First, up to 2000-year different sets of accessions were evaluated at these stations: at DES, practically all new entries; at PEF, accessions primarily from Northern Europe, Canada, and Asian part of Russia; and at YES, entries from south of European Russia, the USA, and Southern Europe. Second, different years were epiphytotic at different stations, and some samples could not be grown at 3 years of high rust development at a certain place, so they were not referred to as resistant at this station. Third, differences in local rust populations for virulence patterns can explain these results.

Since one of the important tasks of VIR is to collect and preserve commercial Russian varieties, the results emphasized the resistance of the varieties from the Wheat Russian Registry. The State Register of Breeding Achievements in 2020 includes 261 varieties of spring bread wheat [50]. Currently, only 15 varieties are missing in the collection. In total, according to results of three epiphytotic years of evaluation, 15, 13, and 8 varieties (see results) were resistant to leaf rust at DES, YES, and PEF, respectively.

According to a widely accepted viewpoint, local varieties (landraces) are of great interest to identify new sources of effective resistance to fungal diseases in cereals [51–53]. They are supposed to have possibly new genes for resistance not used in modern breeding, and theoretically, they could possess the most valuable durable resistance. Among bread wheat landraces from the VIR collection, resistant samples were identified at DES (15) and PEF (10) (see results).

Juvenile (seedling) resistance to the complex population of *P. triticina* was evaluated in 4500 samples, and 73 were classified as resistant. All local samples were susceptible to the

disease, so resistant accessions were presented by varieties and breeding lines, varieties from the Russian Federation being the most presented. Most of these identified samples possess known genes for leaf resistance *Lr9*, *Lr19*, and *Lr24* according to phytopathological tests and molecular DNA marking [4,21,54]. These genes are widely used in the breeding of spring bread wheat in Russia and abroad, and they have lost their effectiveness in some countries and several parts of Russia [23–25].

Evidently, wheat accessions that have been identified as resistant to leaf rust at two to three locations are of greatest interest for breeding purposes. Their characteristics are presented in Table 8.

**Table 8.** Samples of spring wheat resistant to leaf rust in fields of two or more locations.

| VIR Catalogue Number | Variety or Line | Origin | Resistant at Experimental Stations | Types of Reaction in Seedlings | Postulated Effective *Lr* Genes |
|---|---|---|---|---|---|
| 60980 | BR 16—Rio Verde | Brazil | YES, PEF | 3 | |
| 61523 | Henika | Poland | YES, PEF | 3 | |
| 62515 | Amidon | USA | YES, DES | 3 | |
| 62552 | HI 977 | India | YES, DES, PEF | 3 | |
| 62874 | Nordic | USA | YES, DES | 3 | |
| 62875 | Norm | USA | YES, DES | 3 | |
| 62878 | AC Minto | Canada | YES, DES | 3 | |
| 62893 | Ranniaya 93 | Ukraine | YES, DES | 3 | |
| 62898 | CDC Teal | Canada | YES, DES | 3 | |
| 63058 | Fjeld | USA | YES, DES | 3 | |
| 63194 | Swift | Australia | YES, DES | 3 | |
| 63215 | Long 82-2124-1 | China | YES, DES | 3 | |
| 64443 | Tepoca | Mexico | YES, PEF | 3 | |
| 64892 | McKenzie | Canada | YES, PEF | 3 | |
| 64643 | Tulaikovskaya 100 | RF, Samara region | DES, YES | 0 | $Lr6Ag^i2$ * |
| 64647 | Lutescens 30 | RF, Samara region | PEF, YES | 0 | *Lr9* |
| 64666 | Kinelskaya Niva | RF, Samara region | DES, YES | 0 | *Lr19* |
| 64866 | Lubninka | RF, Novosibirsk Region | DES, YES | 0 | *Lr9* |
| 64867 | Novosibirskaya 44 | RF, Novosibirsk Region | DES, YES | 0 | *Lr9* |
| 64872 | Cheliaba Stepnaya | RF, Chelyabinsk Region | DES, YES | 0 | *Lr9* |
| 64997 | Voevoda | RF, Saratov Region | DES, YES | 0 | $Lr6Ag^i1i$ * |
| 64998 | Favorit | RF, Saratov Region | DES, YES | 0 | $Lr6Ag^i1$ * |
| 65000 | Olga | RF, Novosibirsk Region | DES, YES | 0 | *Lr9* |
| 56395 | ANK-4 | RF, Novosibirsk Region | YES, DES, PEF | 0 | *Lr9* |
| 63180 | Tasman | Australia | YES, DES, PEF | 0 | *Lr24* |
| 65816 | LT 1 | RF, Leningrad region | YES, DES, PEF | 0 | |

*—according to Sibikeev et al., 2017; Guiltiaeva, 2018.

Fourteen accessions possess adult resistance, and 12 samples have seedling resistance. All samples with seedling resistance are with one exception line and varieties from the Russian Federation. According to results of PCR amplification with primers specific to

markers of effective genes for leaf rust resistance, 6 accessions have gene *Lr9*, 1 sample is protected by *Lr19*, and 1 sample possesses gene *Lr24* (Figures 7–9, Table 8). Three varieties, Tulaikovskaya 100, Voevoda, and Favorit, possess genes *Lr6Ag^i2* and *Lr6Ag^i1* from *Agropyron intermedium* [21,55], widely used in wheat breeding for rust resistance in the Volga Region of RF. Line LT1 was created at the base of the induction of somaclonal variability in cv. Spica and has genes nonidentical to known effective *Lr* genes.

As a result, wheat samples identified as possessing effective seedling or adult resistance could be of interest for breeding in some regions of the Russian Federation and other countries.

**Author Contributions:** Conceptualization and methodology, Y.V.Z.; and formal analysis, L.G.T. and E.V.Z.; investigation, L.G.T., Y.V.Z., A.N.B., V.A.L., M.A.A., A.Z.S. and E.V.Z.; resources, E.V.Z., A.N.B. and E.Y.K.; data curation, E.V.Z.; original draft preparation, L.G.T.; writing—review and editing, L.G.T. and E.V.Z.; visualization, L.G.T.; project administration, E.V.Z. All authors have read and agreed to the published version of the manuscript.

**Funding:** Ministry of Science and Higher Education of the Russian Federation under Agreement No. 075-15-2020-911, 16 November 2020.

**Data Availability Statement:** Not applicable.

**Acknowledgments:** The article was made with the support of the Ministry of Science and Higher Education of the Russian Federation under Agreement No. 075-15-2020-911, 16 November 2020, providing a grant in the form of subsidies from the federal budget of the Russian Federation. The grant was provided as state support for the creation and development of a world-class scientific center, "Agro-technologies for the Future".

**Conflicts of Interest:** The authors declare no conflict of interest.

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
