# Peer review of "Long-Term Multilocal Monitoring of Leaf Rust Resistance in the Spring Bread Wheat Genetic Resources from Institute of Plant Genetic Resources (VIR)"

_agronomy, doi:10.3390/agronomy12020242_

Round 1
Reviewer 1 Report
Abstract
The abstract is confused and there are no connections between paragraphs explaining results.
Line 27 – to. This paragraph is confused and lack details.
Introduction
The introduction needs more work and information regarding resistance genes, disease cycle, screening methods and types and levels of resistance around the word. Besides that, objectives need more clarification and there are some misspelling words, such as infection and cultivars.
Material and methods
MM also needs some level of work as it does not describe sufficient details to replicate and build on published results. For instance, more information is needed regarding how humidity and temperature affects the disease. Must have a consistence description of temperature among stations. Besides that, there are no description on statistical analysis and randomization of plots and fields.
Line 106 – 113 – Different fond size.
All table and figure headings should be alight to the left as also information, such on table Regions. Besides that, add more space between tables and paragraphs.
Plant Material
Line 122 – Describe field conditions.
Line 127 – How field evaluations were conducted. Add more details.
Line 132 – Same as above.
Methods
Line 140 – Describe conventional agricultural techniques. Where the fields natural infested with inoculum?
Line 141 – 142 – Describe and define optimal time.
Line 150 – Each year?
Line 163 – 165 – Reference and citation of the genes.
Line 184 – Describe at 2-3 experimental stations.
Results
Line 202 – Describe the 34 and 259 samples. Were they inbred lines and etc.
Line 239 – Same as above.
Line 265 – Same as above.
Lines 300 – 329 – Add the list as a table.
Discussion
The discussion has more information than the introduction and some paragraphs should be moved to it.
Overall the manuscript is not well-written in English and lacks a lot of details and information. Although the research provides some level of evidence, the authors still need to clarifying and provide more details about some of the methods, discussion and results.
Author Response
We would like to thank the reviewer for attentive analysis of our article
- Abstract
The abstract is confused and there are no connections between paragraphs explaining results. Line 27 – to. This paragraph is confused and lack details.
We rewrote abstract
- Introduction
The introduction needs more work and information regarding resistance genes, disease cycle, screening methods and types and levels of resistance around the word.
The general task of our work was to give information to breeders, phytopathologists on effective adult and seedling resistance in wheat from one of the biggest Wheat Collection to leaf rust. Information on resistance genes, disease cycle, screening methods and types and levels of resistance is presented in many dozens of reviews and articles. From my personal viewpoint the task of introduction is to designate the problem but not to show the level of knowledge.
Besides that, objectives need more clarification and there are some misspelling words, such as infection and cultivars.
The only objective was to identify stable resistant samples useful for further breeding. Infection was used only once and really, We cannot see the misspelling in its use. Cultivars was replaced be varieties over the text
- Material and methods
MM also needs some level of work as it does not describe sufficient details to replicate and build on published results. For instance, more information is needed regarding how humidity and temperature affects the disease.
It has been described in many reviews and manuals so we did not include this information in our article.
Must have a consistence description of temperature among stations.
We did it
Besides that, there are no description on statistical analysis and randomization of plots and fields.
The field experiments at each station are described in MM. Methods of evaluation are described too. Statistical analysis is needed when we study partial resistance but we identified samples with high level of resistance.
Line 106 – 113 – Different fond size.
It has been corrected
All table and figure headings should be alight to the left as also information, such on table Regions. Besides that, add more space between tables and paragraphs.
It has been done
Plant Material
Line 122 – Describe field conditions.
Temperature and moisture conditions are described. Characteristics of soils were added.
Line 127 – How field evaluations were conducted. Add more details.
The disease ratings were scored twice during the growing season at the heading and milky stages. Classification of a sample for leaf rust resistance was done according to the 1-9 scale
Line 132 – Same as above.
- Methods
Line 140 – Describe conventional techniques. Line 141 – 142 – Describe and define optimal time.
Surу we could describe agricultural techniques of wheat growing and optimal seeding time for each region but from our view point it does not have any relation to the article topic
Where the fields natural infested with inoculum?
We wrote that under field conditions leaf rust evaluation has been done without artificial inoculations
Line 150 – Each year?
Sure
Line 163 – 165 – Reference and citation of the genes.
Is it necessary to describe each genes for resistance if this information is available in many publications?
Line 184 – Describe at 2-3 experimental stations.
Samples identified as resistant only at 1 station were not analyzed with molecular markers. These samples are presented in table 7
5.Results
Line 202 – Describe the 34 and 259 samples. Were they inbred lines and etc.
It has been shown in figure 2 and 3
Line 239 – Same as above.
We added in the text
Line 265 – Same as above.
figure 5 and 6
Lines 300 – 329 – Add the list as a table.
Done
6.Discussion
The discussion has more information than the introduction and some paragraphs should be moved to it.
Discussion of results is more informative than introduction because it compares own results with previously known and introduction poses the problem.
- Overall the manuscript is not well-written in English
It has been checked by native English person.
and lacks a lot of details and information.
What useful information lacks?
Although the research provides some level of evidence, the authors still need to clarifying and provide more details about some of the methods, discussion and results.
We tried do it
Reviewer 2 Report
The manuscript by Tyryshkin Lev Gennadievich et al studied the wheat rust resistance samples from All-Russian Institute of Plant Genetic Resources. Generally speaking, this manuscript integrated previous results from different stations and validated several rust resistance genes in some samples. In my opinion, the manuscript still has lots of room to improve. I listed all my concerns here.
- Many abbreviation names are included in the manuscript, please make sure they are clearly explained. For example, in the title, people will be very confused about the name of VIR, so please add the full name of VIR in the title. In line 61, what does “N.I.” stand for in “N.I? Vavilov All-Russian Institute”?
- The last several sentences in the abstract is very difficult to understand. “fourteen of them possess adult resistance and 12 – _seedling one;” what does “12-_seedling one” mean? I am totally lost about the meaning of “6 accessions have gene Lr9 and 1 – _Lr19 and one – _Lr24”. Please correct the expression of these sentences.
- In the last part of the introduction, a general conclusion should be included.
- I didn’t get the meaning of tables 4, 5, and 6. Why are they important to be included in the manuscript?
- Figures 4, 5, and 7 should be remade to make them easier to read. Please add the title of the Y-axis in all figures. In figure 7, why all country names are abbreviated? It is better to keep consistent with Figures 4 and 5 to use the full name.
- In Figures 8, 9, and 10, the authors detected several rust resistance genes in some samples. The authors' used marker names in the figure legends, for example, in Figure 8, the figure legend is “Amplification products after PCR with primers to SCS5”. It will be better to indicate what resistance gene it is. Because the authors talked about Lr9, Lr19 and Lr24 in the whole manuscript.
- The format of the references should be carefully reorganized, now is pretty inconsistent.
Author Response
We would like to thank the reviewer for attentive analysis of our article
- Generally speaking, this manuscript integrated previous results from different stations
It is not absolutely correct. Most results have never been published. Some results have been published but not in scientific journals, only in VIR catalogues in Russian. And this information is now unavailable for world scientific society. Moreover, we presented here also results on seedling effective resistance
and validated several rust resistance genes in some samples.
Really, we identified effective genes for seedling resistance in all samples that were resistant to the rust in seedlings and simultaneously at 3 experimental stations.
2.Many abbreviation names are included in the manuscript, please make sure they are clearly explained.
We did it.
For example, in the title, people will be very confused about the name of VIR, so please add the full name of VIR in the title.
We did it
In line 61, what does “stand for in “N.I?”?
Our institute officially is called N.I. Vavilov All-Russian Institute of plant genetic resources. N is the first letter of Vavilov’s name, and I is the first letter of Vavilov’s patronymic.
- The last several sentences in the abstract is very difficult to understand. “fourteen of them possess adult resistance and 12 – _seedling one;” what does “12-_seedling one” mean? I am totally lost about the meaning of “6 accessions have gene Lr9 and 1 – _Lr19 and one – _Lr24”. Please correct the expression of these sentences.
We did it.
- In the last part of the introduction, a general conclusion should be included.
We tried to write it
- I didn’t get the meaning of tables 4, 5, and 6. Why are they important to be included in the manuscript?
From our viewpoint, this information is important especially for phytopathologists because it shows epidemiological situation with leaf rust at 3 different regions and its change. We suppose that table presentation is much more evident than text. Also, we add information on evaluation years number in abstract
- Figures 4, 5, and 7 should be remade to make them easier to read. Please add the title of the Y-axis in all figures.
We added Y-axis
In figure 7, why all country names are abbreviated?
We deleted this figure.
It is better to keep consistent with Figures 4 and 5 to use the full name.
Not done because it has been deleted
- In Figures 8, 9, and 10, the authors detected several rust resistance genes in some samples. The authors' used marker names in the figure legends, for example, in Figure 8, the figure legend is “Amplification products after PCR with primers to SCS5”. It will be better to indicate what resistance gene it is. Because the authors talked about Lr9, Lr19 and Lr24 in the whole manuscript.
You are sure right. I add the genes names in figures
- The format of the references should be carefully reorganized, now is pretty inconsistent.
I tried to change the inconsistencies.
Reviewer 3 Report
- A thorough language editing to become more easily readable
- Inclusion of additional relevant and recent literature, not least from the regional area, see for example Rahmatov et al 2019 in Euphytica, screening Tajik wheat material.
- Abstract needs to be revised. The abstract should in principle present the results (50% of text should be results) with only one sentence of introduction, one sentence of materials and methods and a final sentence of conclusion.
Author Response
We would like to thank the reviewer for attentive analysis of our article
- A thorough language editing to become more easily readable
We gave the article to read to 2 native English-speaking people and corrected it according to their remarks
- Inclusion of additional relevant and recent literature, not least from the regional area, see for example Rahmatov et al 2019 in Euphytica, screening Tajik wheat material.
We have really different objectives in our work. In Rahmatov et al 2019 article the study of samples susceptible to diseases was done. Our main objective was identification of stable resistant samples under conditions of Russia that could have practical interest for breeders. So I really don’t understand why it is absolutely necessary to add literature on epidemiological situation in other countries and search of resistant samples in these countries
- Abstract needs to be revised. The abstract should in principle present the results (50% of text should be results) with only one sentence of introduction, one sentence of materials and methods and a final sentence of conclusion.
We revised it.
Round 2
Reviewer 1 Report
Hi Dear,
Thanks for accepting some suggestions.
Reviewer 2 Report
The authors have replied to my concerns properly. I am satisfied with this version.